# Radiation Therapy in the Management of Head and Neck Mucosal Melanoma

**DOI:** 10.3390/cancers16193304

**Published:** 2024-09-27

**Authors:** Omar Azem, Omar Nabulsi, Michael Jelinek, Nikhil Joshi

**Affiliations:** 1Department of Radiation Oncology, Rush University Medical Center, Chicago, IL 60612, USA; nikhil_joshi@rush.edu; 2Rush Medical College, Chicago, IL 60612, USA; omar_nabulsi@rush.edu; 3Department of Internal Medicine, Division of Hematology, Rush University Medical Center, Oncology & Cell Therapy, Chicago, IL 60612, USA; michael_jelinek@rush.edu

**Keywords:** mucosal melanoma, head and neck, postoperative radiotherapy, hypo-fractionation, particle therapy, protons, immunotherapy, sinonasal, oral cavity

## Abstract

**Simple Summary:**

Mucosal melanoma is a rare and aggressive form of melanoma that primarily occurs in the mucosal tissues of the head and neck. Due to its rarity and complexity, treatment strategies for this type of cancer are challenging, with surgical removal often followed by radiation therapy. This review explores how modern radiation therapy techniques, such as intensity modulation, proton beam therapy, and carbon ions, are being used to enhance treatment effectiveness while minimizing damage to surrounding healthy tissues. By reporting on current practices and outcomes, this article aims to provide insights that could guide more effective treatment approaches and further improve outcomes for patients with head and neck mucosal melanoma.

**Abstract:**

Mucosal melanoma of the head and neck (HNMM) is a rare but highly aggressive malignancy, often diagnosed at an advanced stage with poor prognosis. This review discusses current treatment strategies, emphasizing the role of radiotherapy in managing this challenging disease. A comprehensive analysis of 33 studies provides updated information on techniques and outcomes, highlighting the consistent benefit of adjuvant radiation in improving local control. Advances in conformal techniques, such as intensity-modulated radiotherapy (IMRT), have significantly reduced toxicity rates. Preliminary data on proton and carbon ion therapies suggest the potential for further enhancement of the therapeutic ratio, despite limited availability. Although recent studies report 3-year local control rates as high as 90%, overall survival within the same time frame remains well below 50–60%, underscoring the need for continued improvement in systemic therapies to address the persistent issue of distant metastases.

## 1. Introduction

Mucosal melanoma (MM) is a rare entity, comprising one percent of the 100,640 malignant melanomas diagnosed annually, with an incidence of about one per two million [1]. Though most melanocytes exist in the skin, there are limited extracutaneous locations, such as mucosal tissue, where they can be found. Of these mucosal subsites, the highest incidence is in the head and neck (55.4%), followed by the anus/rectum (23.8%), the female reproductive tract (18%), and the urinary tract (2.8%) [2]. Given its low prevalence, definitive risk factors have not been identified. These cancers predominate in older patients, are less promptly detected, often present at an advanced stage, and are associated with a very poor prognosis [3].

## 2. Presentation, Workup, and Staging

Head and neck mucosal melanoma (HNMM) most frequently occurs in sinonasal, oral, or laryngopharyngeal locations^1^. The most common manifestation of MM is a polypoid unilateral lesion with varying degrees of pigmentation (from amelanotic to hyperpigmented), occasionally found in the presence of additional satellite lesions. Though HNMMs are notorious for their asymptomatic and insidious onset, their presentation can involve epistaxis, nasal obstruction, rhinorrhea, headaches, or even dysphagia depending on the location of the lesion [2]. Following a full history and physical examination, complete work-up involves a biopsy, and determination of the local disease extent using cross-sectional imaging (a CT with intravenous contrast and/or an MRI with and without contrast). Evaluation for distant metastasis is essential (a CT of the chest, abdomen, and pelvis with intravenous contrast or a whole-body PET-CT). An MRI of the brain/skull base may be completed if there is concern for the contiguous invasion of the brain.

The eighth edition of the AJCC employs a staging system for HNMM that reflects its generally poor prognosis. The “T” category is limited to T3 (the tumor is limited to mucosa and immediate underlying soft tissue) and T4 (the involvement of deep soft tissue, cartilage, and bone, overlying the skin, brain, skull base, carotid artery, or mediastinal structures) disease. “N” nodal staging consists of N0 (no regional lymph nodes) and N1 (regional lymph node metastases present). “M” metastatic staging consists of M0 (no distant metastases) and M1 (distant metastasis present) [4]. There is no prognostic stage grouping proposed at this time. It is estimated that initial presentation is approximately 70% localized, 20% node positive, and 10% metastatic [5].

## 3. General Treatment Paradigm

In non-metastatic, resectable HNMM, site-specific margin-negative surgical resection is the mainstay of treatment, but is dependent upon the anatomic location and the tumor extent. These surgeries span the spectrum of wide local excision for oral cavity primaries to extensive sinonasal resection and even skull base resection with reconstruction for advanced sinonasal primaries when feasible. Advanced reconstruction can be avoided on a case-by-case basis, especially if this will delay adjuvant radiation (e.g., the reconstruction of a hard palate or maxillectomy defects with free tissue transfer and using an obturator instead). Postoperative radiotherapy is frequently indicated to improve local and regional control. The National Comprehensive Cancer Network (NCCN) guidelines for all HNMMs specifically recommend adjuvant radiotherapy for patients with resected melanoma who exhibit an advanced stage (T4) and node positivity (N1). It is not uncommon for these patients to present with unresectable or medically inoperable diseases. In this subset of patients, definitive RT should be considered to optimize locoregional control [4].

Palliative radiation is often implemented in cases of advanced or metastatic disease, as well as poor performance status. Palliative radiation can target either the primary site/nodes or distant sites, and is useful for symptom relief (e.g., pain control, hemostasis) [4].

The role of systemic therapy in mucosal melanoma is less defined than for its cutaneous counterpart. In certain instances, evidence on chemotherapy for cutaneous melanoma is extrapolated and used for mucosal disease. Compared to cutaneous melanoma, mucosal melanoma generally has lower response rates to immunotherapy in the setting of advanced or metastatic disease [6,7]. Targeted therapies (e.g., BRAF/MEK inhibitors, KIT inhibitors) have less of a role in mucosal melanoma, but these can be utilized as treatment options for patients with select mutations. The role of immunotherapy or targeted therapy in the adjuvant setting is limited due to low or no representation in clinical trials [8,9,10]. The SWOG 1801 and NADINA trials only included a very limited proportion of mucosal melanomas when looking at the role of neoadjuvant pembrolizumab [11]. A phase III trial investigating the role of temozolomide and cisplatin in the adjuvant setting demonstrated a relapse-free survival benefit over interferon, but this regimen is not commonly used in practice [12]. Because of the limited data on systemic treatments in mucosal melanoma, we continue to rely on local treatment therapies.

## 4. Modern Radiotherapy

CT simulation involves making a standard three-to-five-point mask for immobilization, as well as IV contrast and thin-cut slices for optimal anatomic delineation. Notably, simulations in non-reconstructed patients may involve filling the intended cavity (e.g., maxillectomy defect) with water-equivalent material (e.g., a gel-filled condom) for improved dose distribution [13]. An oral stent is useful to separate adjacent anatomical structures (e.g., moving the tongue away from the maxillary sinus/hard palate primary site).

Target volumes have most commonly been limited to the primary site unless the node positivity or oral cavity location prompts the inclusion of regional lymphatics [14]. Primary site radiation typically involves a 2 to 3 cm margin, constrained to the greater anatomic compartment. Elective nodal irradiation is encouraged in the presence of multiple positive nodes, a single node >3 cm, extracapsular extension, or the absence of nodal basin dissection. Gross disease, high risk disease (post-op cavity/neck), and low-risk disease (uninvolved neck) respectively warrant doses of 70 Gy, 60–66 Gy, and 50–60 Gy, all via 1.8–2.0 Gy per fraction. The most common method of radiation delivery is intensity modulated radiotherapy/volumetric modulated arc therapy (IMRT/VMAT) with daily image guidance (cone-beam CT preferred) [4].

Many retrospective HNMM series reveal the consistent utilization of conventional fractionation (~2 Gy per fraction), especially in the adjuvant setting, with a well-established dose response benefit for more than 54 Gy [15]. Figure 1 demonstrates the radiation plan and subsequent treatment response following 66.01 Gy in 35 fractions to a mucosal melanoma of the right ethmoid sinus and ipsilateral neck. In addition to the common conventional schemes (60–70 Gy in 30–35 fractions), hypo-fractionation has been routinely implemented in both the definitive and palliative setting. A notable MD Anderson Cancer Center experience with MMHN utilized a hypo-fractionated regimen of 30 Gy in five fractions for oral cavity primaries, which were deemed to be a safe distance from sensitive CNS structures [16]. Subsequent data identifying improved control rates with the use of 3–4 Gy per fraction have suggested a benefit to increased biologic equivalent doses in this radioresistant histology [17,18]. Though uncommon, there are few reported uses of stereotactic body radiotherapy (SBRT) for sinonasal mucosal melanoma [19,20].

Advanced radiation techniques using particle therapy have become attractive for this disease to improve outcomes and spare critical adjacent structures, thereby limiting treatment morbidity. Proton beam therapy has been increasingly explored due to its potential for sparing critical organs at risk. Similarly, carbon ion therapy is currently under investigation due to its potent radiobiologic effectiveness.

## 5. Updated Results

This narrative review aims to summarize the most recent retrospective studies analyzing HNMM outcomes following radiation (with either photons, protons, or carbon ions). Table 1 and Table 2 present the most updated retrospective reviews conducted internationally across multiple institutions.

Surgery and Post-Operative Radiation (PORT): The treatment modality offering the highest probability of cure in head and neck mucosal melanoma is surgery [21]. The majority of data examining radiation are in the adjuvant setting, since surgery is the preferred initial treatment when possible. One of the largest retrospective HNMM series to date detected a statistically significant 26% improvement in local control with the addition of radiation to surgery [24], consistent with similar reviews [14,31,41,42]. A meta-analysis of over 1500 patients has shown that postoperative radiation can reduce the risk of local recurrence by up to 45% [43]. Though there is a sole meta-analysis showing the potential survival benefit with PORT [44], the body of literature typically demonstrates OS detriment [29] in adjuvant radiation recipients, likely related to the higher tumor stage and adverse margin status of these patients. In recent HNMM trials, with improved patient selection, rates of overall survival with surgery plus radiation are similar to that of surgery alone, as Tsushima et al. achieved a 5-year OS of 46% [45]. Another phase II study of postoperative conventional radiation in 2 Gy per fraction up to 70 Gy, demonstrated a three-year overall survival rate of 44% and local relapse-free survival (LRFS) of 92% [46]. Unfortunately, neither surgery nor surgery plus radiation has had a significant impact upon the rate of distant metastases, which remains the primary cause of death in HNMM [44].

Radiation Alone: There are a few series exploring the utility of definitive radiation for HNMM. In a series of non-surgical patients who received 50–55 Gy in 15–16 Fx, there was an initial complete response in 79%, a local DFS of 49% at three years, and an OS of 17% at five years [36]. A Japanese study of definitive RT obtained similar results with various techniques, such as intraoral mold, intraoral cone electron therapy, interstitial brachytherapy, and external beam radiation (50–76 Gy in 5 to 7 weeks) [47]. Yet another fractionation scheme was observed in this non-operable subset of patients, as Wanger et al. reviewed both twice-daily fractionation at 1.2 Gy per fraction and once-daily fractionation at 1.7 to 3 Gy per fraction, achieving a 1-year local control of 79% and a 5-year OS of 12% [48]. Although Tsushima’s trial was predominantly postoperative, two patients received proton beam therapy via 60 Gy RBE in 15 Fx three times weekly and remained disease-free years later [45]. Additionally, non-curative palliative regimens such as the QUAD-SHOT regimen (14 Gy in four fractions repeated at two-to-three-week intervals for up to three rounds) have proven highly effective for pain control, hemostasis, and tumor shrinkage in patients with locally advanced or metastatic diseases [49].

Particle Therapy: Ramaeker et al. showed increased 5-year local control in sinonasal tumors, favoring protons over photons (88% versus 66%, *p*-value 0.035), though this was in various histologies including, but not specific to, mucosal melanoma [50]. In a 14-patient pilot study treating localized HNMM via hypo-fractionated proton beam therapy, there was a reported three-year local control and OS of 86% and 58%, respectively [33]. A systematic review of 86 observational studies (74 photon, 7 proton, 5 carbon ion) revealed a statistically significant 5-year survival of 44% in patients with HNMM, treated with carbon ions and 25% in those treated with photons [50]. In a group of 72 patients prospectively treated with carbon ions via 52.8 to 64 Gray equivalent in 16 fractions showed 5-year local control, and cancer-specific survival rates of 84% and 40%, respectively [51].

## 6. Reported Toxicities

Some of the more commonly reported acute side effects of radiation in HNMM include dermatitis (dermatitis 13–39%) [15,33], mucositis (3–53%) [15,32], dysphagia (6–25%) [15,35], conjunctivitis, and palpebral edema. Conversely, notable late side effects include vision change/blindness (0–40%) [22,52], xerostomia (6–21%) [14,15], skin necrosis (0–14%) [14,23], brain radionecrosis (6%) [43], and secondary malignancy. In the 2DCRT era, rates of radiation-induced skin necrosis and blindness were reported as high as 14% and 40%, respectively [23,52]. However, in the current era, grade 3 toxicity is less frequent, and grade 4 toxicity is uncommon.

In fact, one of the most recent PORT trials documented no incidences of radiation-induced blindness [46]. In Yao et al., patients were treated with up to 70 Gy in 35 fractions, with grade 3 toxicity rates reported at 12% (one instance of severe dermatitis and one episode of nasal perforation) [46]. Alternative approaches, such as daily hyper-fractionation, have been shown to reduce both acute adverse and late events [23]. Additional modern techniques, such as IMRT and hypo-fractionation, were included in a review from Memorial Sloan Kettering which identified no grade 3 long-term complications [26]. Newer modalities such as proton therapy have demonstrated an improved toxicity profile, with a recently published phase II trial citing no late toxicity of grade 3 or more [33]. Likewise, high linear energy transfer (LET) using carbon ions has also exhibited acceptable toxicity, with no reported grade 3 early or late toxicity in the largest study to date [51].

## 7. Future Directions

Regarding radiation, the standard of care is conventionally fractionated photon therapy in the postoperative setting and hypo-fractionated photon therapy in the definitive setting. IMRT has been heavily adopted due to its enhanced conformality and improved therapeutic ratio. Proton therapy has the potential to further enhance this therapeutic ratio via improved conformality and possible dose escalation. Particle therapy has demonstrated promising efficacy due to its enhanced radiobiologic effectiveness and conformality, but is limited by availability and cost [53]. Ultimately, additional studies are needed to assess the efficacy and toxicity associated with radiation dose escalation/fractionation and particle beam radiation.

Despite advancements in local therapies, the most common pattern of failure in head and neck mucosal melanoma is distant (the lungs, bones, and brain being the most common sites), underlining the importance of systemic therapy. While there is limited data for immunotherapy in mucosal melanoma, checkpoint inhibitors are frequently given in the recurrent/metastatic setting. Whether combination immunotherapy provides increased benefit over single agents in mucosal melanoma is unknown. Targeted therapies may hold some promise for certain individuals, but the detection of signature mutations, such as KIT and BRAF, are reported in under 10% of patients with HNMM [54]. Ongoing work with systemic treatments will be necessary to improve the poor outcomes for patients who develop metastatic disease. There is an ongoing trial investigating the role of nivolumab with cabozantinib in the adjuvant setting, with the hopes of curtailing disease recurrence.

## 8. Conclusions

In conclusion, postoperative radiotherapy has consistently exhibited improved LRC despite its common usage in more locally advanced stages. Even in non-operative cases, radiotherapy is still advised, and should be carried out in a hypo-fractionated, dose-escalated fashion, if feasible. The ability to safely provide more durable/effective definitive treatment has been vastly improved by IMRT. Though photons remain standard, proton beam therapy can be considered for malignant mucosal melanomas when in close proximity to critical structures, such as the eye or in a re-irradiation setting. Likewise, carbon ion therapy has demonstrated favorable experimental results and merits further investigation.

## Figures and Tables

**Figure 1 cancers-16-03304-f001:**
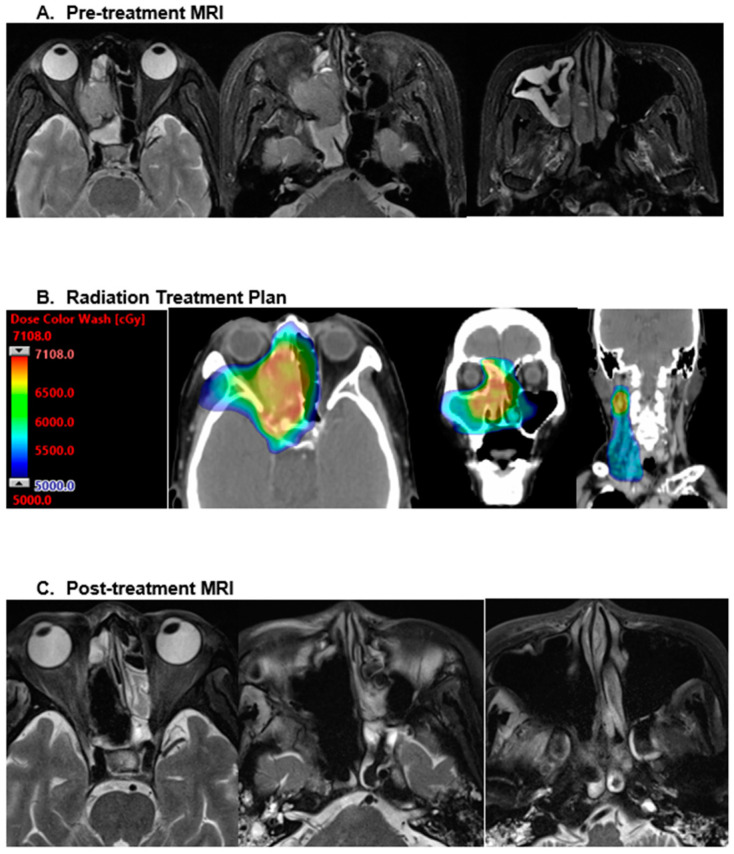
This figure shows the radiation plan and subsequent treatment response following 66.01 Gy in 35 fractions to a mucosal melanoma of the right ethmoid sinus and ipsilateral neck.

**Table 1 cancers-16-03304-t001:** Studies analyzing the outcomes of postoperative radiotherapy for HNMM.

Author	Year	Modality	No. of Patients Receiving RT	Dose/Fx	Dose Range (Gy)	Treatment Outcome (Local Control, Overall Survival)	Toxicity (Acute, Late)
Abt et al. [21]	2021	Photons	437	—	—	Local control: — Survival: 5-year OS of 25%	Acute: — Late: —
Yao et al. [4]	2018	Photons	33 (23 hypo-fractionation CRT and 10 CRT)	2 Gy	65–70 Gy (CTV1) 50–55 Gy (CTV2)	Local control: 92% Survival: 3-year OS of 44.4%	Acute: grade 3, 18% leucopenia Late: grade 3, 12% muscle fibrosis
Frakes et al. [22]	2015	Photons	38	1.5–6 Gy	30–70 Gy	Local control: 3-year LC of 90% Survival: 3-year OS of 59%	Acute: N/A Late: no grade 4 toxicity
Christopherson et al. [23]	2015	Photons, Protons	16	—	—	Local control: 5-year LC of 79% Survival: 5-year OS of 25%	Acute: — Late: grade 4 blindness or skin necrosis, 14%
Thariat et al. [24]	2011	Photons	16	6 Gy	20–60 Gy	Local control: 5-year LC of 49% Survival: 5-year OS of 38%	Acute: radionecrosis 6% Late: none
Vandenhende et al. [25]	2011	—	15	—	—	Local control: 3-year DFS of 38% Survival: 3-year OS of 54%	Acute: — Late: —
Moreno et al. [15]	2010	Photons	33	—	30–66 Gy	Local control: — Survival: —	Acute: grade 2–3 skin toxicity 39% Late: grade 2 mucosal toxicity 6%
Benlyazid et al. [24]	2010	—	78	—	25–70 Gy	Local control: 5-year LRC of 70% Survival: 5-year OS of 27.5%	Acute: — Late: —
Wu et al. [26]	2010	Photons	27	—	—	Local control: 5-year PFS of 22% Survival: 5-year OS of 33%	Acute: grade 2 mucositis 19% Late: no optic complications
Meleti et al. [27].	2008	—	19	—	—	Local control: 26.3% local failure at 9.4 months Survival: —	Acute: — Late: —
Cheng et al. [28].	2007	Photons	12	—	21–60 Gy	Local control: — Survival: —	Acute: — Late: —
Krengli et al. [14]	2006	Photons	42	1.3–5 Gy	5–70 Gy	Local control: 3-year LC of 71% Survival: 5-year OS of 28%	Acute: grade 3 skin and mucosal toxicity 20.8% Late: optic nerve toxicity, stenosis of nasolacrimal duct
Owens et al. [29]	2003	Photons	24	2–6 Gy	30–60 Gy	Local control: 83% LRC Survival: 5-year OS of 29%	Acute: — Late: —
Patel et al. [11]	2002	—	18	—	—	Local control: 5-year local recurrence-free survival 17.7% Survival: 5-year disease-specific survival 39.8%	Acute: — Late: —
Nandapalan et al. [30]	1997	Photons	76	—	—	Local control: — Survival: 10-year OS of 27%	Acute: — Late: —
Temam et al. [31]	1997	Photons	39	2 Gy	50–70 Gy	Local control: LRC of 62% Survival: 5-year OS of 20%	Acute: — Late: —

“—” signifies the absence of the pertinent data point.

**Table 2 cancers-16-03304-t002:** Studies analyzing the outcomes of definitive radiotherapy for intact HNMM.

Author	Year	Modality	No of Patients Receiving RT	Dose/Fx	Dose Range	Treatment Outcome	Toxicity
Abt et al. [21]	2021	Photon	85	-	-	Local control: -	Acute: -
						Survival: 1-year OS of 43.7%, median OS of 10 months	Late: -
Takayasu et al. [32]	2019	Carbon	21	3.5 to 4.5 Gy	57 to 64 Gy	3-year LC: 92%	Acute: G2 mucositis (53%)
						3-year OS: 49%	Late: G3 overall (0%)
Christopherson et al. [23]	2015	Photon and Proton	5	-	66.5–74.4	Local control: -	Acute: -
						Survival: -	Late: bilateral blindness and skin necrosis
Bourgeois et al. [19]	2015	SBRT	2	15 Gy	-	Local control: -	Acute: -
						Survival: -	Late: -
Zenda et al. [33]	2015	Proton	32	4 Gy	60 Gy	Local control: 1-year LC of 76%	Acute: dermatitis grade 3–4 (13%)
						Survival: 3-year OS of 46%	Late: none > grade 3
Greenwalt et al. [34]	2015	Proton	7	-	69.6–74.4	Local control: locoregional control 18 months 86%	Acute: -
						Survival: 18-month OS of 67%	Late: G4 visual toxicity in 14%
Fuji et al. [35]	2014	Proton	20	3.5 Gy	-	Local control: 5-year local control 62%	Acute: mucositis (25%)
						Survival: 5-year OS of 51%	Late: G4 optic nerve toxicity in 15%
Gilligan et al. [36]	2014	-	28	3–3.5 Gy	50–55 Gy	Absolute local control: 61%	Acute: -
						3-year DFS: 49%	Late: -
						5-year OS: 18%	
Ozyigit et al. [20]	2013	SBRT	4	-	-	Local control: -	Acute: -
						Survival: -	Late: -
Shiga et al. [37]	2012	Photon, Proton, Carbon	9	N/A	N/A	Local control: N/A	Acute: -
						Survival: 5-year OS of 30%	Late: -
Zenda et al. [33]	2011	Protons	14	4 Gy per fraction	60 Gy equivalents	3-year LC: 86%	Acute: G3 mucositis (21%)
						3-year OS: 58%	Late: G3 vision changes (14%)
Bachar et al. [38]	2008	Photon	21	-	-	Local control: 5-year LC of 13%	Acute: -
						Survival: 2-year DFS of 26%, median survival of 28 months	Late: -
Combs et al. [39]	2007	Photon IMRT	8	-	60–68 GTV; 54–64 CTV	Local control: -	Acute: skin erythema
						Survival: 5-year OS 80%	Late: no decreased visual acuity
Wada et al. [17]	2003	Photons	31	1.5–13.8 Gy	32–64 Gy (median 50 Gy)	Local control: 61%	Acute: G2 mucositis (16%)
						3-year cause-specific Survival: 33%	Late: death (6.5%)
Nandapalan et al. [30]	2002	Photon	20	-	-	Local control: -	Acute: -
						Survival: 10-year OS of 25%	Late: -
Gaze et al. [40]	1990	Photon	13	-	-	Local control: -	Acute: -
						Survival: -	Late: -

“-” signifies the absence of the pertinent data point.

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
