# Peer review of "Radiation Therapy in the Management of Head and Neck Mucosal Melanoma"

_cancers, 2024, doi:10.3390/cancers16193304_

Round 1

Reviewer 1 Report

Comments and Suggestions for Authors

I found this article on the treatment of mucosal melanoma to be of interest. However, I identify several areas for improvement. Primarily, the header does not include the correct affiliation. Additionally, the figure presented in the abstract is unclear; I am uncertain as to the intended meaning of the term "graphical abstract."

Furthermore, the argumentation lacks a clear basis. It is unclear whether this is a narrative review or if there is a specific methodology guiding the selection of studies. 

It is my opinion that the work is not yet fit for publication, at least in its current form. 

Author Response

Please see attached word document for specific revision acknowledgements. Thank you for your time!

Reviewer 2 Report

Comments and Suggestions for Authors

The authors review radiotherapy for mucosal melanoma, which is considered a rare disease, highlighting recent advances in IMRT and particle therapy, particularly the reduction of doses to organs at risk without compromising dose delivery to the target. The list of articles on radiotherapy for mucosal melanoma would be useful for readers of Cancers.

There are two minor points. 

1) The authors state that evidence on chemotherapy for head and neck mucosal melanoma is scarce because of the rarity of the disease. In practice, evidence on chemotherapy for cutaneous melanoma is extrapolated and used in the treatment of mucosal head and neck melanoma. Therefore, it would be helpful for readers to understand the current approach to chemotherapy if the authors presented those for cutaneous melanoma, such as the NADINA trial presented at ASCO 2024.

2) 4. Modern radiotherapy.l109; the part of the sentence 'doses of 70 Gy, 60-66 Gy and 50-60 Gy' should be accompanied by fractionation.

Author Response

(The authors gave the same response as above.)

Reviewer 3 Report

Comments and Suggestions for Authors

The reviewed manuscript  is a review paper dealing with Head and Neck Mucosal Melanoma. The entity of interest is rare and complex. Therefore an attempt to present common characteristics and risk factors is almost undoable. This is followed by a variety of clinical treatment and its efficiency. Concerning the latter point the authors described modern techniques of radiotherapy followed by two tables containing modality and - among other data - information about such end-points as survival and toxicity. Multiple way of radiotherapy parameters make the tables not easy to understand. Finally, the authors pointed at adjuvant photon-based radiation as the most recommended technique. However this is hardly readable from the text.

Minor remarks

1.      Graphical abstract is  being a combination of Fig 1 A. (left panel) and B (2nd left panel) is lacking any subtitles and as such is unreadable,

2.      There is not given any affiliation of the authors.

3.       There is given (line 46) abbreviation HNMM. It fits well to other abbreviations on the field as HNC or HNSCC.

4.       Explanation of TNM system is commonly known and does not require detailed explanation (lines 58-66). The only difference if relatively large tumors size pointed at T3 and T4.

5.       References  4 and 5 should. Publication year should be inserted just after journal title that is in use in other citations.

Minor correction needed

Author Response

(The authors gave the same response as above.)
